# Cloning and Functional Analysis of *TaWRI1Ls*, the Key Genes for Grain Fatty Acid Synthesis in Bread Wheat

**DOI:** 10.3390/ijms23105293

**Published:** 2022-05-10

**Authors:** Fengping Yang, Guoyu Liu, Ziyan Wu, Dongxue Zhang, Yufeng Zhang, Mingshan You, Baoyun Li, Xiuhai Zhang, Rongqi Liang

**Affiliations:** 1Key Laboratory of Crop Heterosis and Utilization (MOE) and Beijing Key Laboratory of Crop Genetic Improvement, China Agricultural University, Beijing 100193, China; yangfengping@baafs.net.cn (F.Y.); guoyuliu16@163.com (G.L.); wuziyan1125@163.com (Z.W.); zhangdx27@163.com (D.Z.); zhangyufeng@cau.edu.cn (Y.Z.); msyou67@cau.edu.cn (M.Y.); baoyunli@cau.edu.cn (B.L.); 2Beijing Academy of Agriculture and Forestry Sciences, Beijing 100097, China

**Keywords:** bread wheat, *TaWRI1*, fatty acid synthesis, functional analysis, transcription factor

## Abstract

WRINKLED1 (WRI1), an *APETALA2* (AP2) transcription factor (TF), critically regulates the processes related to fatty acid synthesis, storage oil accumulation, and seed development in plants. However, the *WRI1* genes remain unknown in allohexaploid bread wheat (*Triticum aestivum* L.). In this study, based on the sequence of Arabidopsis *AtWRI1*, two *TaWRI1Ls* genes of bread wheat, *TaWRI1L1* and *TaWRI1L2*, were cloned. *TaWRI1L2* was closely related to monocotyledons and clustered in one subgroup with *AtWRI1*, while *TaWRI1L1* was clustered in another subgroup with *AtWRI3* and *AtWRI4*. Both were expressed highly in the developmental grain, subcellular localized in the nucleus, and showed transcriptional activation activity. *TaWRI1L2*, rather than *TaWRI1L1*, promoted oil body accumulation and significantly increased triglyceride (TAG) content in tobacco leaves. Overexpression of *TaWRI1L2* compensated for the functional loss of *AtWRI1* in an Arabidopsis mutant and restored the wild-type phenotypes of seed shape, generation, and fatty acid synthesis and accumulation. Knockout of *TaWRI1L2* reduced grain size, 1000 grain weight, and grain fatty acid synthesis in bread wheat. Conclusively, *TaWRI1L2*, rather than *TaWRI1L1*, was the key transcriptional factor in the regulation of grain fatty acid synthesis in bread wheat. This study lays a foundation for gene regulation and genetic manipulation of fatty acid synthesis in wheat genetic breeding programs.

## 1. Introductions

Bread wheat (*Triticum aestivum* L.) is one of the most important staple crops in China and in the world. Triacylglycerols (TAGs) are one of the main classes of storage components in wheat grains, accounting for about 1.5–3% of the dry weight of the grains [1,2,3]. They are mainly stored in the embryo and aleurone layer and can be extracted from milling byproducts such as the bran and germ of the wheat. TAGs are not only necessary for wheat seed germination, plant growth, and stress resistance [4,5], they are also nutritious for human beings [1].

WRINKLED1 (WRI1), discovered by Focks and Benning in Arabidopsis seed epidermis shrunken mutants, directly regulates fatty acid synthesis and glycolysis in the fatty acid synthesis pathway of plant seeds [6]. As an AP2 gene family member, its protein sequence contains two conserved AP2/ERF domains [7,8,9], and its conserved domain sequences have higher homology with those of the AINTEGUMENTA (ANT)-like group and are therefore considered part of the ANT-like group [7,10].

The *WRI1* transcription factor acts as a regulator and binds to the conserved AW-box [CnTnG(n)CG] element in the proximal upstream region of these genes coding for proteins that function in various metabolic steps of fatty acid synthesis, and directly promotes the expression of these genes [9,11,12,13,14,15]. It is now clear that WRI1 directly binds the proximal upstream region of pyr dehydrogenase (*PDHE1α*), plastidic pyruvate kinase (*Pl-PKpα, Pl-PKpβ1*), ketoacyl-ACP synthases (*KAS1*), acetyl-CoA carboxylase (*BCCP2*), enoyl-acp reductase 1 (*ENR1*), acyl-carrier protein (*ACP1*), and sucrose synthase (*SUS2*), and regulates the expression of these targets [9]. Therefore, *WRI1* is regarded as a distributor of the direction of carbon flow in seeds. *WRI1* also stabilizes root auxin levels by activating the auxin transporter PIN (PIN-FORMEDs) and the auxin-degrading protein Gretchen Hagen 3.3 (GH3.3) [5]. LEAFY COTYLEDON1 and 2 (LEC1 and LEC2) and FUCS3A (FUS3) are upstream regulators of WRI1 [16,17,18,19]; LEC2 directly acts on WRI1 and regulates fatty acid metabolism during seed maturation [16].

In Arabidopsis, three genes, *AtWRI2*, *AtWRI3*, and *AtWRI4*, belonging to the AP2/EREBP family, with high homology to the *AtWRI1* gene, have been cloned [20]. *AtWRI1*, *AtWRI3*, and *AtWRI4* are all involved in triggering acyl chain production, but the expression profiles of these three genes are quite different. Unlike AtWRI3 and AtWRI4, AtWRI1 is mainly responsible for regulating the synthesis of fatty acids in seeds [20]. At present, the *WRI1* genes have been cloned in Arabidopsis, castor, coconut, camellia, corn, Brassica napus, and other plants [6,21,22,23,24,25,26,27]. These *WRI1* genes play a key regulatory role in promoting the synthesis and accumulation of fatty acids in seeds.

Overexpression of *WRI1* was used to drive carbon flux towards fatty acid synthesis in leaves or seeds in some crops, such as tobacco, maize, rice, and camelina [22,24,25,26,27]. Overexpression of exogenous *AtWRI1* in rice not only increased the content of fatty acids in rice seeds but also increased the content of starch in endosperm [26]. Maize *ZmWRI1a* and *ZmWRI1b*, derived from gene replication during evolution, encoded functional proteins and could restore the wrinkle phenotype and low fatty acid content of Arabidopsis *WRI1-4* mutant seeds [27]. The expression of *ZmWRI1* in maize seeds increased the oil content of seeds by 46% compared with the control [22].

The synthesis and accumulation of fatty acids in grains play a crucial role not only in grain germination and seedling morphogenesis [4] but also in human nutrition and grain storage. Unlike dicotyledonous plants such as Arabidopsis, the fatty acids in wheat seeds are mainly accumulated and stored in the embryo. As bread wheat is an allohexaploid crop, the *WRI1* genes may have gene redundancy and functional differentiation. The purpose of this study was to clone the wheat *WRI1* genes from the wheat genome, preliminarily analyze their function, and therefore provide an important reference for further research on the regulation of wheat seed fatty acids to meet genetic breeding objectives.

## 2. Results

### 2.1. Two Orthologs of *AtWRI1*, as Candidate Genes, Were Isolated from Wheat Genome

According to the deduced amino acid sequence of Arabidopsis *AtWRI1* (At3g54320) and by BLASTP alignment with wheat genome, two candidate AP2 genes containing AP2/ERF domain sequences with the highest homology (82.1% and 84.2%, respectively) to *AtWRI1* were screened out; these were the orthologs of *AtWRI1*. These two genes were selected as candidate *TaWRI1Ls* genes in this study, named *TaWRI1L1* (TraesCS5A02G221600, TraesCS5B02G220400, TraesCS5D02G229400) and *TaWRI1L2* (TraesCS5A02G141700, TraesCS5D02G150500), respectively.

Using two pairs of primers (*TaWRI1L1*-F/*TaWRI1L1*-R and TaWRI1L2-F/TaWRI1L2-R) and JSM1 (Jingshengmai1) cDNA, RT-PCR was performed to obtain the target products. The cDNA sequence alignment showed that JSM1*TaWRIL1*-5A, -5B, and -5D were 98.24%, 95.91%, and 100% homologous to TraesCS5A02G221600, TraesCS5B02G220400, and TraesCS5D02G229400, respectively, and the deduced amino acid sequence homologies were 99.24%, 96.49%, and 100%, respectively (Appendix A). JSM1*TaWRI1L2*-5A had the same nucleotide sequence as TraesCS5A01G141700, and JSM1*TaWRI1L2*-5D was 99.77% homologous to TraesCS5D01G150500 (Appendix A). There was no *TaWRI1L2*-5B information in the Chinese Spring genome data, but *TaWRI1L2* (TRIDC5BG025010) from *Triticum dicoccoides* (AABB) has the same sequence as JSM1 *TaWRI1L2*-5B. Taken together, the *TaWRI1Ls* of different wheat varieties had high homology in their nucleotide and deduced amino acid sequences.

The N- and C-terminus sequences of the TaWRI1Ls were quite different. *TaWRI1L1* started with a continuous non-polar amino acid P (Pro), but *TaWRI1L2* was rich in multiple non-polar amino acids A (Aln) at the N-terminus (Appendix A). That could be used as a sequence feature to distinguish these *TaWRI1Ls*. The N-terminus of *TaWRI1L2* was enriched with multiple polar S (Ser) amino acids, which shared the same sequence characteristics with AtWRI1 and WRI1 of other monocotyledonous plants, such as maize and rice, suggesting that TaWRI1L2 may have a similar functional mechanism to that of *AtWRI1*.

The deduced amino acid sequences of eight monocotyledonous, eight dicotyledonous, and six oil plant *WRI1s* were selected to construct an evolutionary tree (Figure 1). TaWRI1L2 was closely related to the monocotyledonous plants’ BdWRI1, ZmWRI1a, ZmWRI1b, and OsWRI1, and clustered into an independent branch. Thus, TaWRI1L2 had the sequence characteristics of monocotyledonous WRI1. However, TaWRIL1 showed higher homology to CeWRI1, AtWRI3, and AtWRI4, and clustered in a subgroup. Therefore, TaWRIL1 may have functions similar to AtWRI3 or AtWRI4.

The bootstrap parameter was 1000.

### 2.2. TaWRI1L1 and TaWRI1L2 Were Predominantly Expressed in the Developmental Grain, Were Subcellular Localized in the Nucleus, and Showed Transcriptional Activation Activity

Using transcriptome data for JSM1 at six grain developmental stages, the expression profiles of *TaWRI1Ls* and 21 paralogs were analyzed. Two homoeologs of *TaWRI1L2* were significantly expressed, and expression of *TaWRI1L2*-5A was the highest, followed by *TaWRI1L2*-5D (Figure 2A). Three homoeologs of *TaWRI1L1* had generally lower expression levels than the two homoeologs of *TaWRI1L2* but the levels were significantly higher than those of the paralogs (Figure 2B). This indicated that *TaWRI1L1* and *TaWRI1L2* play an important role in the development of grain. Based on the expression trend, the expression levels of these two candidate genes decreased continuously with grain development. However, at DAF13, the middle-to-late leaf stage of embryonic development, their expression levels rebounded and appeared as a small peak.

Bars represent the means ± SD of three biological replicates.

Our qPCR results (Figure 2C) revealed that the homologs of *TaWRI1L1* were expressed at three time points (DAF 4, DAF 7, and DAF 10) and showed a downward trend. The homoeologs of *TaWRI1L2* were expressed at all time points, reaching a peak at DAF 7. The homoeologs of *TaWRIL2* were expressed significantly—dozens of times—more than those of *TaWRIL1*. These results indicate that *TaWRIL2* plays a major role in fatty acid synthesis in wheat developmental grains.

The data from the Wheat eFP Browser (Appendix A) also showed that *TaWRI1L1* was mainly expressed in the wheat panicle, flag leaf, glume, and early stage of grain development, while *TaWRI1L2* was mainly expressed in the wheat grain and root tip. *TaWRI1L2* is mainly expressed during embryonic development, while *TaWRIL1* is strongly expressed in endosperm and weakly expressed in embryos. From the perspective of embryonic development, both genes showed peak expression in the leaf late embryo stage, but the expression level of *TaWRIL2* was dozens of times that of *TaWRIL1*.

The transient expression vectors pYBA1132-W1 (*TaWRI1L1*-5B) and pYBA1132-W2 (*TaWRI1L2*-5A) were constructed, respectively (Figure 3A), and used for transient transformation of onion epidermis for subcellular localization of *TaWRI1Ls*. The results (Figure 3B) showed that the expression of both candidate genes was localized in the nucleus of onion epidermal cells, illustrating that both belonged to the nuclear genes and accorded with the typical characteristics of transcription factors.

The pBD-W1 and pBD-W2 yeast transformation experiments (Figure 3C) showed that the yeast cells transformed with *TaWRI1L1*-5B or *TaWRI1L2*-5A genes appeared blue, indicating that the two *TaWRI1Ls* were expressed as fusion genes with GAL4 activity and activated the expression of the reporter gene *LacZ* in yeast AH109. These results confirmed that both *TaWRI1L1* and *TaWRI1L2* genes showed transcriptional activity.

### 2.3. TaWRI1L2, Instead of TaWRI1L1, Promoted Oil Body Accumulation and Significantly Increased Triglyceride Contents in N. benthamiana Leaves

After Agrobacterium GV3101 injection for 5–6 days, the clear outline of *N*. *benthamiana* leaf epidermal cells could be found under a laser confocal microscope (Figure 4A). The obvious accumulation of oil bodies stained with Nile red (indicated by the arrow) could be observed in the transient expressing *TaWRI1L2* leaves, while almost no oil bodies were formed in the control and *TaWRI1L1*-transformed leaves.

To further determine the enhancement of oil body accumulation in *TaWRI1Ls* transient expression leaves, TAG contents in dried leaves were detected with an ORACLE universal fat tester; consistent with the Nile red staining result, expression of *TaWRI1L2* can significantly increase the content of TAG in *N. benthamiana* leaves by 9–10% (Figure 4B). However, transient expression of *TaWRI1L1* could not play an active role in promoting TAG increase in *N. benthamiana* leaves, which was not significantly different from that of the control.

In conclusion, the transient expression results showed that *TaWRI1L2* could positively regulate fatty acid synthesis in *N. benthamiana* leaves, while the regulatory effect of *TaWRI1L1* was not obvious, indicating that the functions of these two *TaWRI1Ls* genes were significantly different.

### 2.4. Overexpression of TaWRI1L2 Compensated for the Functional Loss of *AtWRI1* in an Arabidopsis Mutant and Restored the Wild-Type Phenotypes of Seed Shape, Generation, and Fatty Acid Synthesis and Accumulation

To further confirm the effect of *TaWRI1L2* on fatty acid synthesis, the wild-type Arabidopsis Col-2 and *wri1-1* mutant were transformed with *TaWRI1L2*-5A using the dip flower method. The results showed that the wild-type Col2 seeds were round and plump, and the seed plumpness of Col2-OE-W2 transgenic lines was not significantly different from wild-type Col2 (Figure 5A). The *wri1-1* mutant had shriveled seeds, a shrunken seed coat, and a darker seed coat color than wild-type Col2, while the *wri1-1*-OE-W2 transgenic lines almost recovered the seed shrinkage phenotype of the *wri1-1* mutant and were not significantly different from wild-type Col2 with respect to seed shrinkage, plumpness, and the epidermis (Figure 5A). These results indicated that the expression of the *TaWRI1L2* gene in the *wri1-1* mutant made up for the functional loss of the *WRI1* gene in the mutant and restored the normal level of fatty acid synthesis and accumulation in seeds, thus showing the wild-type seed phenotype.

The transgenic seeds were sterilized and inoculated on MS medium with 3% sucrose to observe the growth of seedlings. The 10-day-old seedlings were collected and stained with Sudan 7B to record the distribution of fatty acids by anatomical microscope. The results showed that wild-type Col2 seeds germinated on the culture medium for 1–2 days and grew into seedlings normally; after Sudan 7B staining, only the hypocotyl and root system of the plant turned light red (Figure 5B). Compared with the Col-2 seedling, the Col2-OE-W2 seedlings grew slightly slower, with expanded cotyledons and hypocotyls, dark color, and relatively strong plants; after Sudan 7B staining, cotyledons, hypocotyls, and the roots of Col2-OE-W2 seedlings were dyed purplish red (Figure 5B), showing that the Col2-OE-W2 seedlings accumulated significantly more fatty acids than wild-type Col2. The *wri1-1*-W2 transgenic seeds germinated slightly more slowly than the wild-type Col-2, and their seedlings grew normally after germination. After Sudan 7B staining, the *wri1-1*-W2 transgenic seedlings had light red hypocotyls and roots and accumulated fatty acids similarly to Col2 (Figure 5B). In a word, the overexpression of the exogenous *TaWRI1L2* gene could make the *wri1-1* mutants restore the wild-type phenotype and significantly improve the utilization of sucrose to activate fatty acid synthesis and accumulation in plants.

Using the seedlings of T_3_ homozygous lines overexpressing *TaWRI1L2* for 10 days after germination, the expression levels of the target genes *Pl-PKβ1*, *BCCP*2, and *KASI* were studied. Each target gene had a significantly higher expression level in Col2-OE-W2 lines than in the wild-type Col2 (Figure 5C), while three target genes expressed in the *wri1-1*-OE-W2 transgenic line could restore the expression level in the *wri1-1* mutant or surpass the expression level in the wild-type Col2 (Figure 5D), indicating that *TaWRI1L2* could activate the three downstream target genes of the *wri1-1* mutant and play a regulatory role in the fatty acid synthesis pathway.

The seeds of the T_3_ homozygous Col2-OE-W2 lines were analyzed for eight fatty acid components (C16:0, C18:0, C18:1, C18:2, C18:3, C20:0, C20:1, and C22: 1). Compared with wild-type Col2, eight fatty acid components in the transgenic lines were increased to different degrees, and the increases in Col2-OE-W2-4 and Col2-OE-W2-9 reached significant levels (Figure 5E,G). Compared with the *wri1-1* mutant, three transgenic lines (*wri1-1*-W2-1, *wri1-1*-W2-2, and *wri1-1*-W2-3) overexpressing *TaWRI1L2* showed increases in eight fatty acid components and total fatty acid content (Figure 5F,H), similar to the content of wild-type Col2. Taken together, overexpression of exogenous *TaWRI1L2* could compensate for the functional loss of *AtWRI1* in mutants and increase the fatty acid content of Arabidopsis seeds.

### 2.5. Knockout ofTaWRI1L2 Reduced the Grain Size, 1000 Grain Weight, and Grain Fatty Acid Synthesis in Bread Wheat

To further study the function of the *TaWRI1L2* gene in wheat, we knocked out this gene using CRISPR. The gRNA sequences for *TaWRI1L2* were screened out on the website https://crispr.bioinfo.nrc.ca/WheatCrispr/ (accessed on 12 November 2018) (Figure 6A), and a vector pBUE411-TaU3p-W2 containing two editing sites was constructed for the transformation of young embryos. Twelve T_0_ plants were positive for the *bar* gene. After two generations of pedigree planting and screening, two homozygous lines, KO-1 and KO-2, derived from two independent T_0_ plants, were obtained. The KO-1 line was edited at the first site by adding a base, and the KO-2 line was edited at the second site by deletion of four bases (Figure 6A,B).

Grains of the KO lines became smaller (Figure 6C) than the receptor Fielder. Meanwhile, grain lengths, grain widths, and 1000 grain weights were also reduced to different degrees (Figure 6E–G). There were no significant differences in imbibition capacity, germination vigor, germination speed (Figure 6D), and developmental process between the KO lines and Fielder.

Five fatty acid components (C16:0, C18:0, C18:1, C18:2, and C18:3) and total fatty acid analyses were performed on the seeds of the KO lines as well as the control Fielder, and the results showed that the contents of oleic acid (C18:1) and linolenic acid (C18:3), were lower in Fielder and not detected in the transgenic lines. The contents of palmitic acid (C16:0) and linoleic acid (C18:2) were significantly lower in KO lines than in Fielder, which resulted in a significant reduction in total fatty acid content in seeds of the two KO lines—only up to 18–20% of the control (Figure 6H). These results illustrated that functional loss of *TaWRI1L2* reduced fatty acid synthesis in wheat grain.

Conclusively, the above functional analyses showed that *TaWRI1L2* had a biological activity and functions similar to the *AtWRI1* gene, such as activating the expression of key genes in the fatty acid synthesis pathway and promoting the synthesis and accumulation of seed fatty acids. The regulatory effect of *TaWRI1L1* is not obvious, indicating that there are significant differences in the functions of the two *TaWRI1Ls* genes.

## 3. Discussion

The fatty acids in plant seeds are essential for seed germination and plant growth [4,5]. Studying the coordination mechanism between the synthesis of fatty acids and the synthesis of starch, protein, and other components in wheat grains could promote directional breeding to meet wheat end-product demand.

Dicotyledonous and monocotyledonous seed structures are distinctly different morphologically due to different endosperm/embryo volume ratios during differentiation. In *Arabidopsis thaliana*, the embryo accumulated lipids and proteins in the cotyledons by depleting nutrients in the endosperm tissue during maturation, and the endosperm cells subsequently degenerated [28]. In cereal crops, embryos are small in size and mainly accumulate lipids and globulins [29], while endosperms persist after cellularization and mainly accumulate starch and storage proteins and undergo programed cell death (PCD) without degradation [30].

Due to obvious differences in the developmental trends of monocotyledonous and dicotyledonous plants, the expression patterns of homologous genes may be different. The expression of *AtWRI1* and its regulated target genes started at 4 days after flowering and then increased rapidly to a maximum at day 8 [6,7,11]. *TaWRIL1* and *TaWRIL2* were expressed at high levels in the developmental grain, while *TaWRIL1* was mainly expressed in the embryo and *TaWRIL2* in the endosperm. However, *TaWRIL2* expression was significantly—dozens of times—higher than that of *TaWRIL1*. *TaWRI1L2* had the sequence characteristics of the monocotyledonous plant *WRI1*, and *TaWRIL1* may have functions similar to *WRI3* or *WRI4*. Therefore, *TaWRI1L2* and its function was further analyzed in tobacco, Arabidopsis, and wheat in this study.

*WRI1* is the regulator of carbohydrate metabolism during the stage of seed filling [6], so its functional loss might affect grain starch synthesis and grain size as well. The Arabidopsis endosperm degenerated and disappeared at the late stage of maturity, and the cotyledons were the main place for oil preservation. The *WRI1* mutation led to a significant decrease in TAG content, so the Arabidopsis seeds shrank significantly [6]. The embryo of bread wheat was the main place for oil storage and could not cause obvious grain shrinkage when grain fatty acid synthesis was affected. Therefore, grain shrinkage was not the phenotype of *TaWRI1*. We found that there was no difference in germination and subsequent growth and development between the KO grains and wild-type grains, probably because the starch stored in the endosperm was the major carbon source and energy provider during the germination stage. The overexpression of *AtWRI1* in rice can lead to an increase in starch content and a significant decrease in fatty acid content in *WRI1* rice endosperm [26]. In our study, the knockout of *TaWRI1L2* in wheat resulted in a significant decrease in grain size and weight. Thus, it can be seen that *WRI1* acts in different ways in various plant and organs or tissues, the mechanisms of which need to be further explored in future research.

Maize *Z**mWRI1a* and *Z**mWRI1b*, derived from gene replication during evolution, shared 85.32% amino acid sequence similarity [27]. *Z**mWRI1a* was mainly expressed in the maize embryo and had two expression peaks at the early stage of grain development and at the end of the grain filling stage [27], which was consistent with the expression pattern of *TaWRI1L2*. *Z**mWRI1b* was highly expressed in the endosperm and had a peak during grain development [27] which resembled the expression pattern of *TaWRI1L1*. However, *TaWRI1L1* and *TaWRI1L2* had 38.64% amino acid sequence similarity, and *TaWRI1L2* had higher homology with *OsWRI1* and *Z**mWRI1* than *TaWRI1L1*. Therefore, *TaWRI1L2*, along with *OsWRI1* and *Z**mWRI1*, had a functional mechanism similar to *AtWRI1*.

The expression of *Z**mWRI1* increased the oil content of seeds by 46% without affecting seed germination or plant growth, and there was no significant difference in yields between transgenic lines and control lines [22]. The C16 and C18 fatty acid contents of maize seeds were significantly increased in the lines overexpressing *Z**mWRI1a* compared to the control [27]. Our results showed that *TaWRI1L2*-5A was more expressed than *TaWRI1L2*-5B and *TaWRI1L2*-5D in wheat seeds, and that its over-expression increased oil production and accumulation in tobacco leaves, while its knockout reduced the oil contents in wheat seeds. Therefore, we can manipulate the key regulator *TaWRI1L2*-5A in future breeding programs to regulate TAG synthesis to produce more vegetable oil in modified biomass for increased oil nutritional value or less oil for the longer storage life of wheat seeds.

## 4. Materials and Methods

### 4.1. Cloning and Sequence Analysis of Target TaWRI1Ls Gene

Using the induced amino acid sequence of the AP2 protein domain of the Arabidopsis thaliana *AtWRI1* gene (At3g54320), BLASTP alignment was performed on the Chinese Spring genome data using the website database at http://plants.ensembl.org/index.html (accessed on 16 April 2017) to screen out the homology genes. According to the orothlog data of the gene provided by the database, many candidate *TaWRI1Ls* (TaWRI1-likes) genes containing AP2/ERF domain sequences were found in the wheat genome. Two pairs of primers, *TaWRI1L1*-F/ ORF-*TaWRI1L1*-R and *TaWRI1L2*-F/*TaWRI1L2*-R, were designed in the conserved regions at the 5’ and 3’ ends of *TaWRI1Ls* to amplify the ORF region (Table 1).

The bread wheat variety “Jingshengmai1 (JSM1)”, bred by the Beijing Agricultural Biotechnology Research Center, was used to extract total RNA from grains 4–22 days after flowering (DAF) for RT-PCR amplification and RNA-sequence analysis.

The evolutionary relationship between *TaWRI1Ls* candidate genes and *WRI1* genes of model plants (such as Arabidopsis thaliana, maize, and cabbage) was analyzed with MEGA7.0 (Mega Limited, Auckland, New Zealand). The evolutionary tree was constructed using the neighbor-joining method (NJ), and the bootstrap parameter was 1000.

### 4.2. Expression Analysis of TaWRI1Ls

According to the description of the morphological and developmental stages of wheat grains [31], the grains of “Jingshengmai 1” was collected at 4DAF (pre-embryo), 6DAF (transition), 10DAF (leaf early), 13DAF (leaf mid), 16DAF (leaf late), and 19DAF (mature) for qPCR and RNA-seq expression analysis. The expression of *TaWRI1Ls* in JSM1 grains was investigated by qPCR using an SYBR Premix ExTaq Kit (Takara Branch, Beijing, China) with three biological replicates. The relative expression levels were evaluated by the 2^−ΔΔCt^ method [32].

The expression data for the *TaWRI1Ls* gene were downloaded from the Wheat eFP Browser (http://bar.utoronto.ca/efp_wheat/cgi-bin/efpWeb.cgi (accessed on 8 May 2017)) for expression profile analysis [33].

### 4.3. Subcellular Localization and Yeast Activation Analysis of TaWRI1Ls

The ORF fragments of *TaWRI1L1*-5B (TraesCS5B02G220400) and *TaWRI1L2*-5A (TraesCS5A02G14170) of the candidate *TaWRI1Ls* gene (with restriction sites) were cloned into the downstream of CaMV 35S promoter in the GFP fusion vector pYBA1132, and the recombinant vectors, named pYBA1132-W1 and pYBA1132-W2 (Figure 3A), respectively, were used for subcellular localization by transient expression. The onion epidermis was bombarded with a BIO-RAD high-pressure helium PDS1000 gene gun (BIO-RAD Co., Hercules, CA, USA), and the subcellular localization of the target genes was detected by confocal laser observation 24 h later.

The ORF fragments of the *TaWRI1L1*-5B and *TaWRI1L2*-5A genes were cloned into the yeast BD expression vectors and the recombinant vectors, named pBD-W1 and pBD-W2, were introduced into the yeast strain AH109 to observe the color of the yeast cells after staining with X-gal to determine whether the target genes had transcriptional activation activity.

### 4.4. Functional Analysis of TaWRI1Ls in N. benthamiana Leaves

The *TaWRI1L1*-5B and *TaWRI1L2*-5A fragments were respectively constructed into the overexpression pYBA1302 vector containing the herbicide selection marker, and the recombinant vectors pYBA1302-W1 and pYBA1302-W2 were respectively transferred into Agrobacterium strain GV3101. Taking the Agrobacterium containing the empty vector as the control, the Agrobacterium solution containing the recombinant vector was injected into the back of *N. benthamiana* leaves at the same position.

After 3–4 days, oil accumulation could be observed in injected leaves. After 6–7 days, the injected leaves were harvested for intracellular oil analysis. The oil bodies inside the cells were observed by means of a laser scanning microscope (red fluorescence at wavelengths 560 nm and 615 nm) after the leaves were stained with 10 μg/mL Nile red solution (Sigma-N3013) for 30 min at room temperature and washed with 0.1 M Tris-HCl buffer (pH 8) for 10 min [24].

Triglyceride (TAG) contents in tobacco leaves were detected with an ORACLE universal fat tester (CEM Co., Matthews, NC, USA). One gram of dried leaves of each sample was measured with three biological replicates to calculate the percentage (*w*/*w*) of TAGs in the dry leaves.

### 4.5. Functional Analysis of TaWRI1Ls in Arabidopsis

The wild-type *Arabidopsis thaliana* Columbia 2 (Col2) and its *wri1-1* mutant (obtained from the Arabidopsis Information Resource) were used as receptors, respectively, and the transformation was performed by the floral dip method using the Agrobacterium strain GV3101 containing the overexpression vectors pYBA1302-W1 and pYBA1302-W2. The transgenic Arabidopsis seeds were sterilized and sown on ½ MS solid medium with 3% sucrose. One week after germination, the seedlings were observed and stained with Sudan Red 7B to observe the degree of oil accumulation. Red staining indicates the accumulation of neutral lipids in Arabidopsis seedlings [18].

After harvest, the content of each fatty acid component in the seeds was determined with three biological replicates by gas chromatography [34], according to the manufacturer’s instructions.

Three target genes, *Pl-PKβ1* (At5g52920), *BCCP2* (At5g15530), and *KASI* (At5g46290), were selected [9] to analyze the regulatory effects of the overexpressed *TaWRI1L2* gene. The sequences of the qPCR primer were shown in Table 1.

### 4.6. Functional Analysis of TaWRI1Ls in Bread Wheat

The gRNA fragment of the *TaWRI1L2*-5A gene was screened on the website https://crispr.bioinfo.nrc.ca/WheatCrispr/ (accessed on 12 November 2018) and used to construct the CRISPR vector pBUE411-TaU3p-W2. The spring wheat cultivar “Fielder” embryos were transformed with Agrobacterium strain EHA105 containing this CRISPR vector, and the transgenic plants were obtained after callus induction, callus differentiation, and plant regeneration.

T_0_–T_2_ transgenic plants and their recipient variety Fielder were grown in the greenhouse. The leaves of the transgenic plants were used to extract genomic DNA. The PCR was performed using specific PCR primers and the high-fidelity Taq PCR mix (SuperStar Co., Beijing, China ), and the target fragments separated by 1% agarose gel electrophoresis were sequenced (Shanghai Shenggong Co., Shanghai, China) to determine the editing sites.

After harvest, the contents of each fatty acid component in the seeds were measured as mentioned above. The grain lengths, grain widths and 1000 grain weights of Fielder and the knockout lines were recorded with three biological replicates using the rapid SC-G grain appearance quality image analysis system (Hangzhou WSeen Detection Technology Co., Ltd., Hangzhou, China) [35].

### 4.7. Statistical Analysis

Means and standard deviations were calculated for each treatment and the statistical differences were analyzed by Student’s *t*-test at *p* < 0.05 using SPSS 16.0 (SPSS, Inc., Chicago, IL, USA).

## Figures and Tables

**Figure 1 ijms-23-05293-f001:**
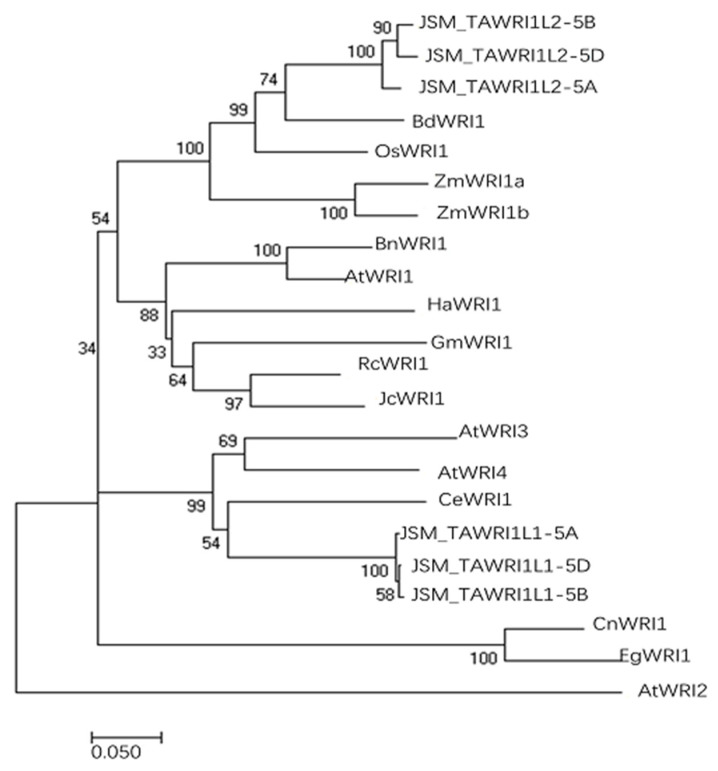
The evolutionary tree of *TaWRI1Ls* and *WRI1s* of other plants.

**Figure 2 ijms-23-05293-f002:**
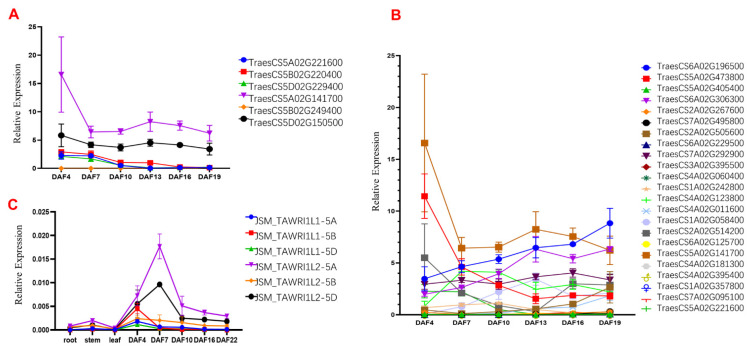
Expression levels of *TaWRI1Ls* and their paralogs in Chinese Spring and Fielder. (**A**) *TaWRI1Ls* of Chinese Spring grains at different filling stages; (**B**) the 21 paralogs of Chinese Spring grains at different filling stages; (**C**) *TaWRI1Ls* of Fielder root, stem, leaf, and grain at different filling.

**Figure 3 ijms-23-05293-f003:**
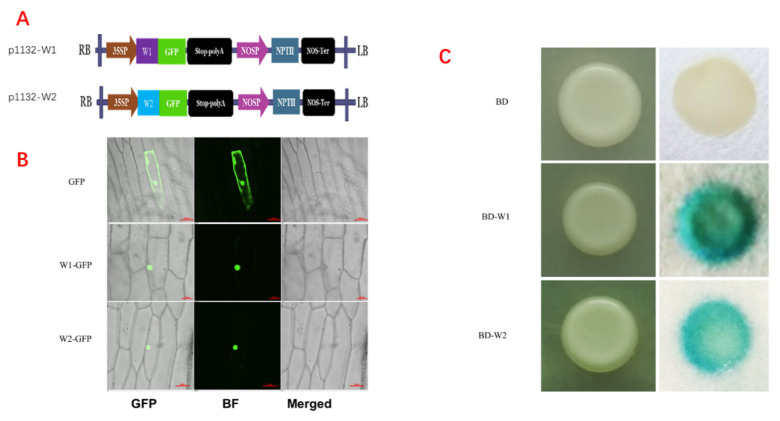
Expression characteristics of *TaWRI1Ls*. (**A**) The transient expression vectors pYBA1132-W1 (*TaWRI1L1-5*B) and pYBA1132-W2 (*TaWRI1L2*-5A). (**B**) The subcellular localization of *TaWRI1Ls*. Bars = 50 µm. (**C**) Yeast transcriptional activity analysis of pBD-W1 (*TaWRI1L1*-5B) and pBD-W2 (*TaWRI1L2*-5A).

**Figure 4 ijms-23-05293-f004:**
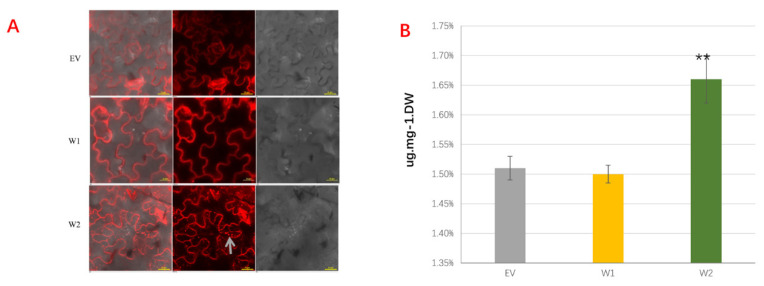
Functional analyses of *TaWRI1Ls* in *N. benthamiana* leaves. (**A**) The oil bodies under the laser confocal microscope. Arrow indicate oil bodys. (**B**) The triglyceride contents in tobacco leaves. EV: empty vector; W1: pYBA1302-W1 (*TaWRI1L1*-5B); W2: pYBA1302-W2 (*TaWRI1L2*-5A). Data are represented as means ± SD from independent biological replicates, and *p*-values are indicated by Student’s *t*-test (** *p* < 0.01). Bars = 25 µm.

**Figure 5 ijms-23-05293-f005:**
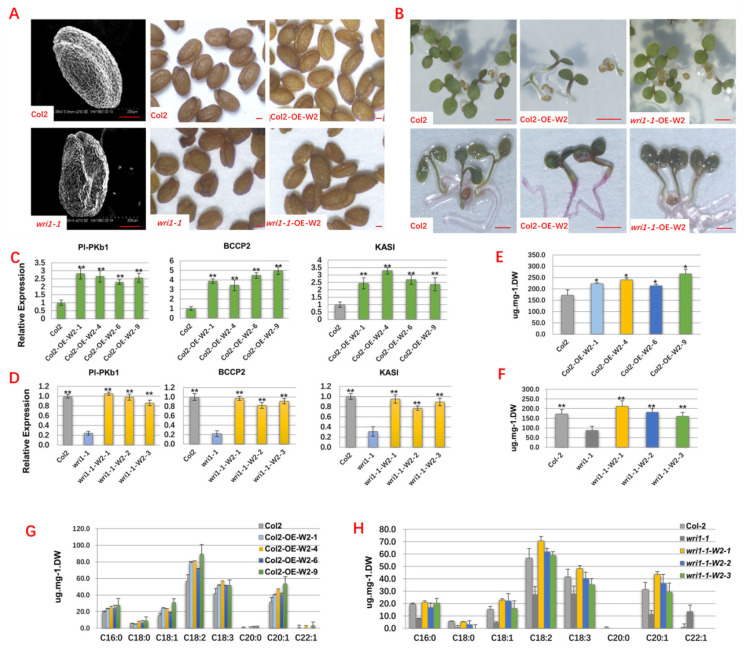
Functional analyses of *TaWRI1Ls* in the *Arabidopsis* mutant. (**A**) Seeds of Col2, *wri1-1* mutant, and their respective OE lines. Bar = 100 µm. (**B**) Upper: seedlings of Col2, *wri1-1* mutant, and their respective OE lines; lower: seedlings of Col2, *wri1-1* mutant, and their respective OE lines stained with Sudan Red 7B. Bar = 500 mm. (**C**) Expression of *Pl-PKβ1*, *BCCP2*, and *KASI* in Col2 and its OE lines. (**D**) Expression of *Pl-PKβ1*, *BCCP2*, and *KASI* in *wri1-1* and its OE lines. (**E**) Seed fatty acid contents of Col2 and its OE lines. (**F**) Seed fatty acid contents of *wri1-1* and its OE lines. (**G**) Seed fatty acid components of Col2 and its OE lines. (**H**) Seed fatty acid components of *wri1-1* and its OE lines. Data are represented as means ± SD from independent biological replicates, and *p*-values are indicated by Student’s *t*-test (* *p* < 0.05, ** *p* < 0.01).

**Figure 6 ijms-23-05293-f006:**
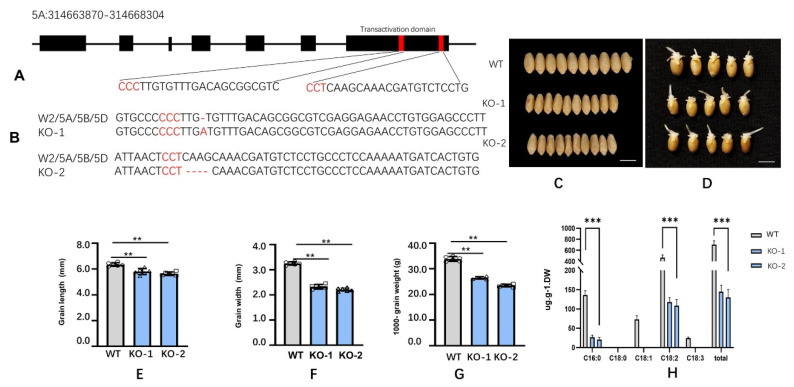
Phenotypes of wheat *TaWRI1L2* knock-out mutants. (**A**) The location of gRNA. (**B**) The edited sites of the KO mutants. (**C**) Mature grains and their generation of WT and KO mutants. Bars = 5 mm. (**D**) Generation of WT and KO mutants. Bars = 5 mm. (**E**) The grain lengths of WT and KO mutants. *n* = 3. (**F**) The grain widths of WT and KO mutants. *n* = 3. (**G**) The 1000 grain weight of WT and KO mutants. *n* = 3. (**H**) Grain fatty acid components of WT and its KO mutants. Data in (**C**,**D**) are represented as means ± SD from independent biological replicates, and *p*-values are indicated by Student’s *t*-test (** *p* < 0.01, *** *p*< 0.001).

**Table 1 ijms-23-05293-t001:** The primers for RT-PCR and qPCR.

Primer Names	Sequence (5′–3′)	Length of Product
*TaWRI1L1*-F	CAGCAGCGCAATGGCAAAG	
*TaWRI1L1*-R	TCACAAGTCCAGCTCGAA	1204 bp
*TaWRI1L2*-F	CCACCATTGACACAGCACAG	
*TaWRI1L2*-R	AAACCTGAAACCTTCCTTGG	1352 bp
Pl-PKb1-F	CCATATGAGTGAGATGTTTGC	
Pl-PKb1-R	AAGGACGATAGTGACTTAAC	119 bp
BCCP2-F	GCAGCTCGACTGTGAGATCG	
BCCP2-R	GTCTGCCATTACAGGAGGCAT	100 bp
KAS1-F	ACAAGCTGTGGACTTTGACAC	
KAS1-R	TGAAGGCAGAGAAGGCGACTAC	119 bp

## Data Availability

Data supporting this work are available within the paper and its Appendix A. RNA-seq data were uploaded to the NCBI with the accession number PRJNA791126.

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
