# Peer review of "Cloning and Functional Analysis of TaWRI1Ls, the Key Genes for Grain Fatty Acid Synthesis in Bread Wheat"

_ijms, 2022, doi:10.3390/ijms23105293_

Round 1
Reviewer 1 Report
The Manuscript entitled “Cloning and functional analysis of TaWRI1Ls, the key genes 2 for grain fatty acid synthesis in bread wheat” is importance for gene regulations and genetic manipulation of fatty acid synthesis 24 in wheat genetic breeding programs. This paper is no doubt emphasize on advance tools of molecular techniques aiming for finding fatty acid synthesis and their metabolic pathways related for crop improvement.
However, I find few points needs to be addressed for improving this manuscript-
- Introduction section WRI1 genes should be included;
- Methodology and Result sections only molecular experimentations are not sufficient. Screening of morphological features along with relevant images are essential following molecular data;
- Discussion sections must be improved with suitable examples of functions of WRI1 genes in crop plants;
- Put some discussion what in the practical implementation of this knowledge acquired from these studies, i.e., Translational point of view.
Considering the aforementioned flaws reviewer suggest for major revision and resubmission of this manuscript.
Author Response
Dear Reviewer 1,
Thank you very much for your comments on our manuscript, and your suggestions greatly help us improve our research. We have checked the manuscript and revised it according to your valuable comments.
Please find the responses here, and revised manuscript in attachment.
Point 1: Introduction section WRI1 genes should be included;
Response 1: We agree with this suggestion and add some advance in this section, from line 64 to line 71. In fact, cloning and features of WRI1 were introduced in the previous version, and we really hope that this section was improved.
Point 2: Methodology and Result sections only molecular experimentations are not sufficient. Screening of morphological features along with relevant images are essential following molecular data;
Response 2: We ae happy to edit the text further, based on helpful comments from the reviewers. From line 428 to line 432, is about the screening methods for morphological features of transgenic wheat lines.
We agree that the data of morphological features along with relevant images are essential. In fact, there are some description and images of morphology in result 2.3, 2.4 and 2.5. We think we listed the necessary description and images.
Point 3: Discussion sections must be improved with suitable examples of functions of WRI1 genes in crop plants;
Response 3: Based on helpful comments from the reviewers, we add two samples in this section and in introduction section. We will edit the text further.
Point 4: Put some discussion what in the practical implementation of this knowledge acquired from these studies, i.e., Translational point of view.
Response 4: This is a valuable suggestion. We discuss the practical implementation of other crops, and predict the prospects in wheat breeding.
Other revision:
1、we add ** to the figure 5 and 6.
2、we add 2 References, and move the position of 1 reference.
3、in Methods section, we change “standard errors” into “standard deviations”.
4、Minor modification:several words and punctuation markers.
Reviewer 2 Report
Soft wheat (Triticum aestivum L.) is one of the main agricultural crops in the world. The study of the inheritance of traits of grain quality in soft wheat is of particular interest to breeders. Therefore, the presented article is relevant and has undoubted scientific value and interest for researchers.
The manuscript contains up-to-date information and is well structured.
The literary review is quite complete, modern. 9 works for the last 5 years are given.
The manuscript is scientifically substantiated, the experimental design is suitable for testing the hypothesis.
The results of the manuscript are reproducible, the section "Methods" is presented quite fully.
Drawings, images, and diagrams display data correctly and are easy to interpret and understand.
The authors carried out statistical processing of the data. Means and standard errors were counted for each treatment and the statistical differ-367 ences were analyzed by t-test at P< 0.05 using SPSS 16.0 (SPSS, Inc., Chicago, IL).
The conclusions made in the work are consistent with the evidence and arguments presented.
I believe that the submitted manuscript can be accepted for publication in its present form.
Author Response
Dear Reviewer 2,
Thank you very much for your comments on our manuscript, and your sugestions greatly help us improve our research. We have checked the manuscript and revised it according to your valuable comments.
Please find the responses in attachment file.
Thank you for your time and consideration.
